# UnLoc: Leveraging Depth Uncertainties for Floorplan Localization

**Matthias Wüest[1,2], Francis Engelmann[3,4], Ondrej Miksik[5], Marc Pollefeys[1,5], Daniel Barath[1,6]**

[1]ETH Zurich   [2]ZHAW   [3]Stanford University   [4]USI Lugano   [5]Microsoft   [6]HUN-REN SZTAKI

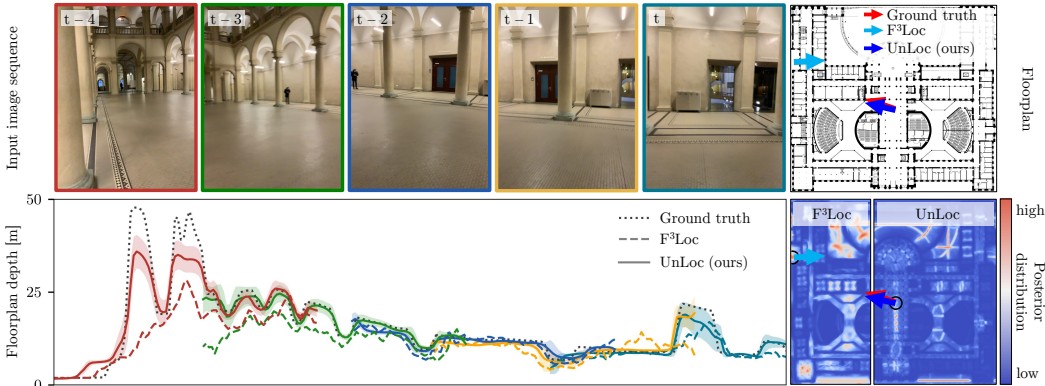

Figure 1: UnLoc processes an input image sequence to predict the floorplan depth (in meters) and associated uncertainty for each image column. Using these predictions, it generates a probability distribution over potential SE(2) camera poses and outputs the most likely one (blue arrow). The ground truth pose is also shown (red arrow), overlapped by the predicted pose.

## Abstract

We propose UnLoc, an efficient data-driven solution for sequential camera localization within floorplans. Floorplan data is readily available, long-term persistent, and robust to changes in visual appearance. We address key limitations of recent methods, such as the lack of uncertainty modeling in depth predictions and the necessity for custom depth networks trained for each environment. We introduce a novel probabilistic model that incorporates uncertainty estimation, modeling depth predictions as explicit probability distributions. By leveraging off-the-shelf pre-trained monocular depth models, we eliminate the need to rely on per-environment-trained depth networks, enhancing generalization to unseen spaces. We evaluate UnLoc on large-scale synthetic and real-world datasets, demonstrating significant improvements over existing methods in terms of accuracy and robustness. Notably, we achieve 2.7 times higher localization recall on long sequences (100 frames) and 42.2 times higher on short ones (15 frames) than the state of the art on the challenging LaMAR HGE dataset. Code and materials: https://github.com/matthias-wueest/UnLoc

## 1 Introduction

Camera localization within indoor environments is a fundamental problem in computer vision, essential for applications in augmented reality and robotics. Accurate localization enables devices to understand their spatial context, facilitating tasks such as navigation, object interaction, and autonomous exploration. Traditional localization methods often rely on pre-built 3D models (Sattler et al., 2011; 2016b; Liu et al., 2017; Sarlin et al., 2019; Panek et al., 2022) or extensive image databases (Schonberger & Frahm, 2016; Arandjelovic et al., 2016; Keetha et al., 2023; Wei et al., 2024), which are storage-intensive and require substantial maintenance, limiting their scalability to new or dynamically changing environments.

Floorplans offer a lightweight and readily available alternative for indoor localization. As 2D representations of spaces, floorplans are easy to obtain (Liu et al., 2018; Yue et al., 2023) and remain unaffected by changes in appearance, such as furniture rearrangements or lighting variations. Recent methods have leveraged floorplans for localization by aligning images with the map (Howard-Jenkins & Prisacariu, 2022; Min et al., 2022; Chen et al., 2024a), enabling devices to localize within new environments as long as a floorplan is available.

Among these, $F^3$Loc (Chen et al., 2024a) has emerged as a recent promising approach for sequential visual floorplan localization, significantly outperforming all previous baselines. It integrates observations over time using a histogram filter, achieving impressive accuracy and computational efficiency on a range of datasets. However, as promising as it is, $F^3$Loc has notable limitations that hinder its practical deployment, which we aim to improve upon in this paper:

**Lack of Uncertainty Modeling.** $F^3$Loc combines monocular and multi-view depth estimation. However, it assumes that the resulting predictions are all of similar accuracy, and it has no means to represent and predict the uncertainties in the depth predictions. In indoor environments, depth estimation is often unreliable in regions with glass walls, open doorways, or large, featureless walls. When fusing a sequence of predictions, not accounting for uncertainty, inaccurate depth predictions adversely affect the localization process, leading to erroneous pose estimates.

**Dataset-Specific Depth.** $F^3$Loc relies on a custom depth prediction network trained separately for each dataset or environment. This per-dataset training requirement poses significant challenges for scalability and robustness. Collecting sufficient depth data for retraining the depth network for every new environment is impractical, especially when rapid deployment is desired.

This paper addresses these shortcomings by introducing a novel visual floorplan localization method that incorporates uncertainty estimation into monocular depth prediction and uses it to robustly fuse predictions from a sequence of images. Also, our method enables us to leverage pre-trained monocular depth models. Our contributions are as follows:

*Uncertainty-Aware Depth Prediction:* We model floorplan depth predictions as explicit probability distributions, assuming a Laplace distribution centered at the predicted depth with scale parameter given by the uncertainty estimate. This formulation allows us to represent the confidence associated with each prediction. This uncertainty-aware approach improves localization by weighting predictions according to their reliability in challenging regions, and it further provides principled weights for the post-processing optimization, yielding higher accuracy.

*Leveraging Off-the-Shelf Depth Models:* Rather than designing or training custom depth networks for each environment as $F^3$Loc does, we directly employ state-of-the-art monodepth models pre-trained on large-scale datasets (Yang et al., 2024b). Our formulation treats these models as plug-and-play modules, demonstrating that reliable localization can be achieved with any sufficiently strong depth predictor, without requiring environment-specific retraining.

The proposed UnLoc (see Fig. 1) achieves significant improvements in accuracy and robustness across multiple datasets compared to prior methods.

## 2 RELATED WORK

**Visual localization** is a fundamental problem in computer vision, addressed through various approaches. Traditional methods include image retrieval techniques (Chum et al., 2007; Jégou et al., 2010; Arandjelovic et al., 2016; Keetha et al., 2023; Wei et al., 2024), which find the most similar images in a database and estimate the pose of the query image based on the retrieved ones. Structure-from-Motion-based approaches (Agarwal et al., 2011; Schonberger & Frahm, 2016; Sattler et al., 2016a; Panek et al., 2022) build a 3D model of the environment and establish 2D-3D correspondences by matching local descriptors, computing camera poses using minimal solvers (Kukelova et al., 2008) and RANSAC (Fischler & Bolles, 1981) or its recent variants (Barath et al., 2020; Barath & Matas, 2021). Scene coordinate regression methods (Brachmann et al., 2017a;b) learn to regress the 3D coordinates of image pixels, while pose regression techniques (Kendall et al., 2015; Kendall & Cipolla, 2017) use networks to predict a 6-DoF camera pose from input images directly. More recently, we saw methods that combine language-based retrieval (Chen et al., 2024b) in combination with scene graph representations (Miao et al., 2024; Zhang et al., 2025). These methods often

rely on pre-built 3D models that are storage-intensive and scene-specific, limiting their applicability in unseen environments.

To overcome this, **floorplan-based localization** methods have emerged, utilizing overhead images or floorplans to estimate the SE(2) pose of the camera (Workman et al., 2015; Tian et al., 2017). These approaches can localize images in new scenes as long as a floorplan is provided. Floorplan localization is frequently associated with LiDAR sensors (Mostofi et al., 2014; Yin et al., 2019; Zimmerman et al., 2022), which are impractical for widespread mobile device use. Alternative methods reconstruct 3D geometry using depth cameras (Winterhalter et al., 2015) or visual odometry (Mur-Artal et al., 2015). Some approaches extract geometric features like room edges to align with the floorplan (Boniardi et al., 2019; Lin et al., 2019). However, these methods often assume known camera or room height, which is not always feasible.

Recent learning-based methods aim to use only RGB images for floorplan localization. Orienter-Net (Sarlin et al., 2023) localizes images in 2D public maps such as OpenStreetMap using neural matching, but focuses on outdoor environments. LaLaLoc (Howard-Jenkins et al., 2021) estimates the position of panoramic images in a floorplan by embedding map and image features into a shared space. LaLaLoc++ (Howard-Jenkins & Prisacariu, 2022) removes the known camera and ceiling height assumption by directly embedding the floorplan. LASER (Min et al., 2022) represents the floorplan as a set of points and uses PointNet (Qi et al., 2017) to embed the visible points for each pose, aligning them with image features in a shared space. PF-Net (Karkus et al., 2018) integrates localization within a differentiable particle filtering framework, using a learned similarity between images and corresponding map patches. F$^3$Loc (Chen et al., 2024a) utilizes metric monocular depth prediction but requires training custom depth networks, which can limit generalizability, and assumes that all predictions are of the same quality.

**Sequence-based localization** methods enhance robustness by integrating information over time. Bayesian filtering (Dellaert et al., 1999; Chu et al., 2015; Karkus et al., 2018; Boniardi et al., 2019; Mendez et al., 2020) is commonly used to fuse sequential observations, like particle (Dellaert et al., 1999) and histogram filters (Thrun, 2002). PF-Net (Karkus et al., 2018) employs a differentiable particle filter for localization but relies on learned observation models that may not generalize well. F$^3$Loc (Chen et al., 2024a) uses sequential observations and integrates them by a histogram filter.

**Depth estimation** provides valuable geometric information for localization (Chen et al., 2024a). Recent advances in pre-trained monocular depth (monodepth) models (Ranftl et al., 2021; Birkl et al., 2023; Guizilini et al., 2023; Yin et al., 2023; Hu et al., 2024; Yang et al., 2024a;b; Bochkovskii et al., 2024), trained on large-scale datasets, enable accurate metric or relative depth predictions out-of-the-box, without per-scene training. In this work, we leverage such off-the-shelf networks for image-based floorplan localization, thereby avoiding custom model training and improving generalization across domains.

**Depth uncertainty** is crucial for reliable use of depth predictions in downstream tasks. Neural network uncertainty is commonly categorized as aleatoric (data) or epistemic (model) (Kendall & Gal, 2017; Poggi et al., 2020). Aleatoric uncertainty captures inherent observation noise by modeling a distribution over the network output. It is most relevant in regions of the observation space with higher noise and does not decrease with more data. A typical approach trains the network to predict parameters of a parametric distribution via log-likelihood maximization (Nix & Weigend, 1994), usually adding negligible computational overhead (Kendall & Gal, 2017).

Epistemic uncertainty reflects model limitations by placing a distribution over the model parameters. It is most useful with small datasets or safety-critical tasks and decreases as more data becomes available. Estimating epistemic uncertainty generally incurs significant computation cost (Kendall & Gal, 2017). Common strategies include Monte Carlo Dropout (Srivastava et al., 2014), bootstrapped ensembles (Lakshminarayanan et al., 2017), and Bayesian neural networks (MacKay, 1992).

For pixel-wise monocular depth estimation, prior work has explored both types. Kendall & Gal (2017) propose a method to estimate aleatoric and epistemic uncertainty jointly. Poggi et al. (2020) evaluate approaches of both types and propose a self-teaching model for aleatoric uncertainty. Liu et al. (2019) model aleatoric uncertainty by discretizing depth into bins. Roessle et al. (2022) predict aleatoric uncertainty via Gaussian depth distributions with learned variance. In floorplan localization, F$^3$Loc (Chen et al., 2024a) estimates floorplan depth assuming a uniform confidence. In contrast, we model aleatoric uncertainty explicitly to capture ambiguities from glass walls, door-

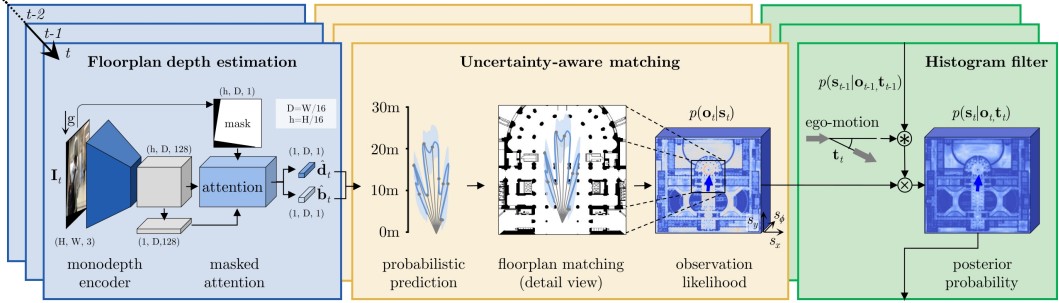

Figure 2: **Main method overview.** At timestep $t$, UnLoc aligns an image with gravity and processes it through a monodepth encoder. The extracted features, along with a binary mask from the gravity alignment, are used to predict the floorplan depth $\hat{\mathbf{d}}_t$ and uncertainty $\hat{\mathbf{b}}_t$ via masked attentions. These predictions form equiangular rays, allowing for uncertainty-aware matching with the floorplan's occupancy map. A histogram filter fuses the observation likelihood with the integrated past belief.

ways, and occlusions. Although epistemic uncertainty could help for out-of-distribution scenes, we focus on aleatoric uncertainty for computational efficiency in real-time histogram filtering, leaving epistemic modeling for future work. Our Laplace-based formulation enables uncertainty-aware matching, improving robustness and convergence in challenging indoor environments.

## 3 FLOORPLAN LOCALIZATION WITH UNLOC

**3.1. Method Overview.** We estimate the 2D pose $\mathbf{s}_t = [s_{x,t}, s_{y,t}, s_{\phi,t}]$ within a known 2D floorplan, where $s_{x,t}$ and $s_{y,t}$ denote position coordinates and $s_{\phi,t}$ represents orientation. Our approach takes as input a sequence of $t$ RGB images, relative poses between these images, gravity direction, camera intrinsics, and the geometric layout of the floorplan. To ensure out-of-the-box usability of the method, input floorplans are provided solely as occupancy grid data without any semantic labels.

An overview of our main method is shown in Fig. 2. Each image is aligned with gravity and processed by a pre-trained encoder (Yang et al., 2024b) to extract features. These features, along with a binary mask from the gravity alignment, are input to a masked attention mechanism that predicts the floorplan depth, defined as the depth to the nearest occupied area in the floorplan. The predicted depth is then used to construct equiangular rays, which are matched against the floorplan to produce a localization estimate. Finally, we combine this estimate with the previous posterior localization using a histogram filter to compute the updated posterior estimate.

Our approach differs from prior methods in two main ways: (1) we model floorplan depth as an explicit probability distribution to quantify uncertainty, enhancing sequential localization when fusing predictions; (2) we use an off-the-shelf pre-trained monodepth network instead of a custom network that requires per-dataset training, thereby effectively leveraging models trained on extensive data without additional training efforts.

**3.2. Gravity Alignment.** Real-world applications often involve images captured from hand-held or head-mounted devices, resulting in arbitrary orientations. To address this, we preprocess images to align them with gravity. We achieve alignment by utilizing the camera's roll ($\psi$) and pitch ($\theta$) angles to define rotation matrices. The rotation for roll is given as $\mathbf{R}_x(\psi)$ and for pitch as $\mathbf{R}_y(\theta)$. The combined matrix from the gravity-aligned frame to the camera frame is calculated as $\mathbf{R}_{cg} = \mathbf{R}_y(\theta) \cdot \mathbf{R}_x(\psi)$. The inverse transformation, from the camera frame to the gravity-aligned frame, is given by: $\mathbf{R}_{gc} = \mathbf{R}_{cg}^\top$. Using these rotations, we compute a homography $\mathbf{H}$ to warp the image into the gravity-aligned frame $\mathbf{H} = \mathbf{K} \cdot \mathbf{R}_{gc} \cdot \mathbf{K}^{-1}$, where $\mathbf{K}$ is the camera intrinsic matrix. This process also produces a binary mask indicating pixels that are invalid after alignment due to the warping. The gravity-aligned RGB image and the mask are then used for subsequent feature extraction.

Note that the gravity direction and camera intrinsics are usually accessible from sensors on smartphones and head-mounted devices. If this information is not directly available, methods such as GeoCalib (Veicht et al., 2024) can be used to estimate both the intrinsics and gravity direction.

**3.3. Feature Extraction.** Recent visual floorplan localization methods (Howard-Jenkins et al., 2021; Howard-Jenkins & Prisacariu, 2022; Min et al., 2022; Chen et al., 2024a) rely on encoders pre-trained on ImageNet (Deng et al., 2009) for image classification tasks, such as ResNet-50 (He et al., 2016). However, we posit that encoders trained on tasks more closely related to floorplan depth estimation could yield better performance. To explore this, we utilize encoders pre-trained on dense monocular depth estimation tasks. These encoders, optimized on large-scale depth datasets, provide superior features for floorplan depth estimation without requiring additional depth training from scratch. From the state-of-the-art models (Birkl et al., 2023; Guizilini et al., 2023; Yin et al., 2023; Hu et al., 2024; Yang et al., 2024a;b; Bochkovskii et al., 2024), we select the recent Depth Anything v2 model (Yang et al., 2024b) due to its state-of-the-art performance in both relative and metric depth estimation and low inference times compared to similarly accurate models (Bochkovskii et al., 2024). Specifically, we use the encoder fine-tuned for indoor environments. We extract features from its last layer and apply bilinear interpolation to match the spatial dimensions the subsequent masked attention mechanism requires. Note that the proposed pipeline is *agnostic* to the monodepth network, which could easily be replaced as better methods are published.

**3.4. Masked Attention.** We implement a masked attention mechanism inspired by Chen et al. (2024a) to predict floorplan depth, using the interpolated features and gravity alignment mask as inputs. Our model predicts two 1D vectors: $\hat{\mathbf{d}}_t$, representing floorplan depth estimates, and $\hat{\mathbf{b}}_t$, denoting the uncertainty associated with each estimate. The uncertainty quantification allows the model to account for varying confidence levels across regions in the image, particularly in challenging areas with ambiguous visual cues or distant objects.

Before inputting to the attention mechanism, we reduce the channel dimensions of the interpolated encoder features using a convolutional layer. The resulting features serve as keys and values in the attention mechanism, while 1D queries are formed through average pooling. Positional encodings for the queries are derived from their 1D coordinates, whereas for the keys and values, positional encodings are mapped from the corresponding 2D image coordinates. By applying the gravity alignment mask, we focus the attention mechanism on observable regions of the image.

The output of the masked attention layer is fed into two parallel fully connected layers: one predicting the depth estimates $\hat{\mathbf{d}}_t$, and the other predicting the uncertainties $\hat{\mathbf{b}}_t$. This dual output allows for uncertainty-aware matching with the floorplan in subsequent steps.

**3.5. Uncertainty-Aware Matching.** Using the predicted depth $\hat{\mathbf{d}}_t$ and uncertainty $\hat{\mathbf{b}}_t$, we compute the observation likelihood over the entire floorplan. The predicted uncertainty $\hat{\mathbf{b}}_t$ represents aleatoric uncertainty, capturing the inherent observation noise in floorplan depth estimation that arises from scene ambiguities such as glass surfaces, open doorways, and featureless walls. To formulate the observation model, we treat each predicted depth value as drawn from a probability distribution. We model the predicted floorplan depth and its uncertainty as our observation $\mathbf{o}_t$ and define the observation likelihood as

$$p(\mathbf{o}_t \mid \mathbf{s}_t) = \prod_{j=1}^{R} \frac{1}{2 \cdot \tilde{b}_{t,j}} \cdot \exp\left( -\frac{|\tilde{d}_{t,j} - d_j(\mathbf{s}_t)|}{\tilde{b}_{t,j}} \right), \tag{1}$$

where $\tilde{d}_{t,j}$ and $\tilde{b}_{t,j}$ are the predicted depth and uncertainty interpolated from $\hat{\mathbf{d}}_t$ and $\hat{\mathbf{b}}_t$ at ray angle $\alpha_j$, and $R$ is the number of rays. The corresponding floorplan depth $d_j(\mathbf{s}_t)$ is computed from the floorplan ray length as

$$d_j(\mathbf{s}_t) = r_j(\mathbf{s}_t) \cdot \cos(\alpha_j), \tag{2}$$

where $r_j(\mathbf{s}_t)$ is the ray length from pose $\mathbf{s}_t$ in direction $\alpha_j$. In Eq. 1, the observation likelihood is modeled as a product of independent Laplace distributions, with $\tilde{d}_{t,j}$ as location parameter and $\tilde{b}_{t,j}$ as scale parameter. We choose the Laplace distribution for two key reasons. First, its heavier tails compared to the Gaussian distribution provide robustness to larger prediction errors that commonly occur in challenging indoor scenes (see Sec. A.2.4 for empirical evidence). Second, it enables efficient closed-form likelihood computation essential for real-time histogram filtering. When uncertainty $\tilde{b}_{t,j}$ is high, the distribution becomes flatter, naturally down-weighting unreliable observations in the pose estimation process. As a result, the filter can rely more on confident predictions while remaining robust to uncertain ones. This uncertainty-aware matching yields a 3D likelihood volume representing the observation likelihoods of all possible camera poses $\mathbf{s}_t$.

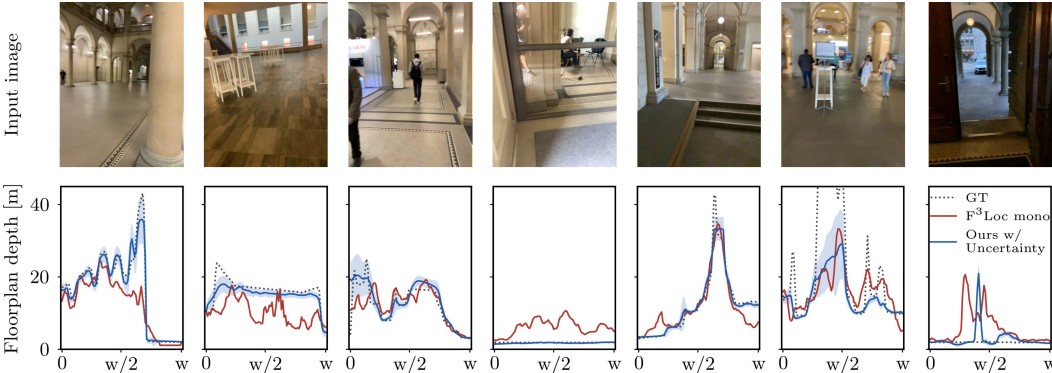

Figure 3: **Floorplan depth predictions** (in meters) for images from the LaMAR HGE dataset. **Top**: input images. **Bottom**: depth predictions by F$^3$Loc (red) and our proposed UnLoc (blue), with predicted uncertainties visualized. The horizontal axis represents the image column index, ranging from left (0) to right (image width $w$). A gray dotted line indicates the ground truth depth.

| Model | T=100, N=37 SR@1m (%) ↑ | RMSE (succ.) ↓ | RMSE (all) ↓ | T=50, N=78 SR@1m (%) ↑ | T=35, N=111 @1m (%) ↑ | T=20, N=200 @1m (%) ↑ | T=15, N=268 @1m (%) ↑ |
|---|---|---|---|---|---|---|---|
| GT Depth | 100.0 | 0.07 | 0.07 | 98.7 | 91.0 | 76.0 | 72.0 |
| LASER | 59.5 | 0.39 | 1.96 | – | – | – | – |
| F$^3$Loc fusion | 94.6 | 0.12 | 0.51 | 94.6 | 69.4 | 46.0 | 41.8 |
| F$^3$Loc mono | 89.2 | 0.18 | 0.88 | 70.5 | 55.9 | 34.0 | 28.4 |
| F$^3$Loc mono* | 86.5 | 0.14 | 0.80 | 70.5 | 57.7 | 35.5 | 32.1 |
| + Depth Anything v2 | 94.6 | **0.11** | 0.50 | 89.7 | 76.6 | 60.5 | 56.3 |
| **UnLoc** w/o post-processing | **97.3** | 0.16 | **0.28** | 92.3 | 88.3 | 70.5 | 65.3 |
| **UnLoc** | **97.3** | 0.16 | **0.28** | 94.9 | 92.8 | 86.5 | 81.3 |

Table 1: **Sequential localization** on the Gibson(t) dataset (Xia et al., 2018). We report the success rates and the RMSE over the successful and all sequences when the sequence length (T) is 100. We report the success rate for all other lengths, considering a localization a success if the accuracy of the last 10 frames is within 1m of the ground truth (GT). We also show the number of sequences tested (N) in each setting. In the first row, we report the localization accuracy with the GT depth. * indicates that the F$^3$Loc model was trained by us.

**3.6. Histogram Filter.** We estimate the posterior probability of the pose over time using a histogram filter, similar to the one introduced by Chen et al. (2024a). At each time step, we update the posterior by combining the likelihood with the prior belief propagated through the motion model. To do this, the posterior is formulated as a 3D probability volume and expressed via Bayes' theorem as

$$p(\mathbf{s}_t \mid \mathbf{o}_t, \mathbf{t}_t) = \frac{1}{Z} \sum_{\mathbf{s}_t} p(\mathbf{s}_t \mid \mathbf{s}_{t-1}, \mathbf{t}_t) \cdot p(\mathbf{o}_t \mid \mathbf{s}_t), \tag{3}$$

where $Z$ is a normalization constant, $\mathbf{t}_t = [t_{x,t}, t_{y,t}, t_{\phi,t}]$ is the vector of ego-motion measurements, and $p(\mathbf{s}_t \mid \mathbf{s}_{t-1}, \mathbf{t}_t)$ is the transition probability. Let us highlight that the uncertainty $\hat{\mathbf{b}}_t$ of the depth prediction directly affects Eq. 3 through its last element: $p(\mathbf{o}_t \mid \mathbf{s}_t)$ (see Eq. 1).

The motion model describes the evolution of the state given the ego-motion measurements and is defined as follows:

$$\mathbf{s}_t = \mathbf{s}_{t-1} \oplus \mathbf{t}_t + \boldsymbol{\omega}_t, \tag{4}$$

where $\boldsymbol{\omega}_t = [\omega_{x,t}, \omega_{y,t}, \omega_{\phi,t}]$ represents Gaussian transition noise with covariance $\boldsymbol{\Sigma} = \mathrm{diag}(\sigma_x^2, \sigma_y^2, \sigma_\phi^2)$, and $\oplus$ denotes the state update operation. The transition probability can thus be written as

$$p(\mathbf{s}_t \mid \mathbf{s}_{t-1}, \mathbf{t}_t) = \exp\left(-\frac{1}{2}(\mathbf{s}_t - \mathbf{s}_{t-1} \oplus \mathbf{t}_t)^\top \boldsymbol{\Sigma}^{-1}(\mathbf{s}_t - \mathbf{s}_{t-1} \oplus \mathbf{t}_t)\right).$$

To efficiently implement the update, the multiplication with the transition probability in Eq. 3 is handled using two decoupled filters: translation and rotation. See Chen et al. (2024a) for additional

details on the filter implementation. The resulting probability volume corresponds to the posterior probability. By sequentially applying the transition and observation updates, we obtain the posterior probability distribution over the camera pose at each time step.

**3.7. Training.** We train our model by minimizing the negative log-likelihood of the predicted depth and uncertainty with respect to the ground truth (GT) floorplan depth at the GT pose. Specifically, we use the following loss:

$$L_d = \sum_{i=1}^{D} \left( \log(\hat{b}_i) + \frac{|\hat{d}_i - d_i(\mathbf{s})|}{\hat{b}_i} \right), \tag{5}$$

where $\hat{d}_i$ and $\hat{b}_i$ are the predicted depth and uncertainty for the $i^{\text{th}}$ image column, $d_i(\mathbf{s})$ is the GT floorplan depth at the GT pose $\mathbf{s}$, and $D$ is the number of image columns for which depth is predicted. This loss corresponds to the negative log-likelihood of a Laplace distribution, encouraging accurate depth predictions while accounting for uncertainty. The logarithmic term discourages infinite predictions for uncertainty.

**3.8. Post-Processing Optimization.** While our main method provides accurate camera poses, residual drift and misalignment with local depth measurements may accumulate. To reduce such errors, we perform a lightweight post-processing optimization over the last $k$ frames. We use $k = 10$ in our experiments. The goal is to refine the trajectory by applying a global rigid correction, parameterized by an $\text{SE}(2)$ transformation composed of an in-plane rotation and translation. We initialize the trajectory by computing the last pose as $\hat{\mathbf{s}}_T = \arg\max_{\mathbf{s}_T} p(\mathbf{s}_T \mid \mathbf{o}_T, \mathbf{t}_T)$ and then backpropagating $k - 1$ steps using the ego-motion measurements and the inverse motion model without noise as

$$\hat{\mathbf{s}}_t = \hat{\mathbf{s}}_{t+1} \ominus \mathbf{t}_{t+1}, \quad t \in [T - k + 1, T - 1], \tag{6}$$

where $\ominus$ is the inverse state update operation. Let the resulting, estimated trajectory over the last $k$ frames be denoted as $\{\hat{\mathbf{s}}_t\}_{t=T-k+1}^{T}$. We introduce a global $\text{SE}(2)$ correction $\Delta\mathbf{s}$ acting on the XY-plane of the camera poses, parameterized by angle $\theta \in \mathbb{R}$ and translation $\mathbf{p} \in \mathbb{R}^2$ as follows:

$$\tilde{\mathbf{s}}_t(\theta, \mathbf{p}) = \Delta\mathbf{s}(\theta, \mathbf{p}) \cdot \hat{\mathbf{s}}_t, \quad t \in [T - k + 1, T]. \tag{7}$$

The optimization minimizes the uncertainty-weighted sum of differences between predicted depths $\tilde{d}_{t,j}$ and floorplan depths $d_{t,j}$ at the refined poses $\tilde{\mathbf{s}}_t(\theta, \mathbf{p})$ across all rays of the last $k$ frames as

$$\mathcal{L}_{\text{post}}(\theta, \mathbf{p}) = \sum_{t=T-k+1}^{T} \sum_{j}^{R} \frac{1}{\tilde{b}_{t,j}} \cdot |\tilde{d}_{t,j} - d_j(\tilde{\mathbf{s}}_t(\theta, \mathbf{p}))|. \tag{8}$$

This weighted L1 objective is designed so that frames with higher predicted uncertainty contribute less to the optimization, which is expected to help the refinement remain robust to noisy predictions. The use of an $\text{SE}(2)$ correction keeps the refinement computationally lightweight while enforcing global consistency in the local window of frames. We solve for the optimal correction parameters $(\theta^*, \mathbf{p}^*) = \arg\min_{\theta, \mathbf{p}} \mathcal{L}_{\text{post}}(\theta, \mathbf{p})$ using a gradient-based optimizer, with floorplan depths $d_j(\tilde{\mathbf{s}}_t(\theta, \mathbf{p}))$ updated at each iteration.

## 4 EXPERIMENTS

**Datasets.** We evaluate our method and the baselines on three datasets: Gibson (Xia et al., 2018), and two versions of LaMAR (Sarlin et al., 2022). The *Gibson* dataset (Xia et al., 2018) contains 118 synthetic scenes (each smaller than 300m$^2$). Following Chen et al. (2024a), we use two subsets: Gibson (f) for training (24,779 sequences of 4 frames each), and Gibson (t) for testing (118 trajectories ranging from 280 to 5,152 steps). It features upright camera poses, low to medium occlusion, and a large FoV of $108°$.

*LaMAR:* To evaluate in real-world settings, we use a subset of LaMAR (Sarlin et al., 2022), focusing on indoor sequences of the HGE building. This large-scale scene covers approx. 22,500 m$^2$. The dataset includes camera poses, degrees of occlusion from low to high, and a narrow FoV of $48°$. It consists of 16 sessions totaling 5,187 images, split into 12 sessions for training (3,820 images), one for validation, and 3 for testing. While prior work (Chen et al., 2024a) is only evaluated on a custom version of this scene, cropping challenging parts (e.g., long corridors), we use the entire scene.

| | Model | SR@1m (%) ↑ | T=100, N=11 RMSE (succ.) ↓ | RMSE (all) ↓ | T=50, N=24 SR@1m (%) ↑ | T=35, N=35 @1m (%) ↑ | T=20, N=63 @1m (%) ↑ | T=15, N=85 @1m (%) ↑ |
|---|---|---|---|---|---|---|---|---|
| **Original** | GT Depth | 100.0 | 0.20 | 0.20 | 91.7 | 85.7 | 73.1 | 56.5 |
| | F³Loc mono | 36.4 | 0.45 | 27.38 | 16.7 | 5.7 | 1.6 | 1.2 |
| | + Depth Anything v2 | **100.0** | 0.38 | 0.38 | 66.7 | 42.9 | 23.8 | 9.4 |
| | UnLoc w/o post-processing | **100.0** | 0.34 | 0.34 | **75.0** | 60.0 | 36.5 | 20.0 |
| | **UnLoc** | **100.0** | **0.25** | **0.25** | **75.0** | **74.3** | **63.5** | **50.6** |
| **Cropped** | GT Depth | 100.0 | 0.23 | 0.23 | 100.0 | 96.2 | 76.1 | 58.1 |
| | F³Loc mono | 75.0 | 0.53 | 4.93 | 29.4 | 7.7 | 2.2 | 0.0 |
| | + Depth Anything v2 | **100.0** | 0.51 | 0.51 | 82.4 | 53.9 | 34.8 | 16.1 |
| | UnLoc w/o post-processing | **100.0** | 0.45 | 0.45 | **94.1** | 76.9 | 50.0 | 33.9 |
| | **UnLoc** | **100.0** | **0.41** | **0.41** | **94.1** | **88.5** | **71.7** | **62.9** |

Table 2: **Sequential localization** on the LaMAR HGE dataset and on its cropped version used by Chen et al. (2024a). We report the success rates (SR) and RMSE over the successful and all sequences when the sequence length (T) is 100, and the SR for all other lengths, considering a localization a success if the accuracy of the last 10 frames is within 1m of the GT. We also show the number of sequences tested (N). The first row reports the localization accuracy with the GT depth.

| | Gibson(t) | | LaMAR HGE | |
|---|---|---|---|---|
| Model | Extraction | Matching | Extraction | Matching |
| F³Loc fusion | 0.030s | 0.003s | – | – |
| F³Loc mono | 0.015s | 0.003s | – | – |
| F³Loc mono* | 0.015s | 0.003s | 0.089s | 0.797s |
| + Depth Anything v2 | 0.185s | 0.003s | 0.179s | 0.746s |
| **UnLoc** | 0.174s | 0.004s | 0.185s | 0.880s |

Table 3: **Runtime** in secs on the Gibson (Xia et al., 2018) and LaMAR HGE datasets (Sarlin et al., 2022). We independently show the depth prediction time per frame and match it to the floorplan.

**Baseline.** We compare to F³Loc (Chen et al., 2024a) as it is the state-of-the-art floorplan localization approach. We use their pre-trained models and also train them ourselves with the provided code to make them applicable to the LaMAR dataset. While we consider its purely monocular version (F³Loc mono) as our main competitor, we also show results for F³Loc fusion, which uses both monodepth and multiview stereo. Let us note that our method can also easily benefit from multiview stereo. Additionally, to study the individual contribution of our uncertainty-aware depth prediction, we evaluate our approach without post-processing.

**Metrics.** We report the success rate (SR) and consider sequential localization at $X$ meters successful if the prediction is within a radius of $X$ meters over the last 10 frames. We compare SR for various numbers of frames. Also, we show the RMSE (over the last 10 frames) of our trajectory tracking in both succeeded and all runs.

**Results on Gibson(t).** Table 1 reports results on the synthetic Gibson(t) dataset. We evaluate SR (%) and RMSE (in meters) over both successful and all sequences for length-100 sequences, and report SR for other lengths, considering localization successful if the accuracy over the last 10 frames is within 1 meter of the ground truth (GT). The number of sequences is also shown. The first row gives the upper bound performance using GT depth. Note that post-processing can still improve upon it.

Without post-processing, our method already achieves substantial gains over both the original F³Loc and its variant using DepthAnything v2 for depth prediction, thanks to uncertainty modeling. Post-processing further boosts performance, especially on short sequences. For instance, on 15-frame sequences, it yields a 16-point SR improvement, even surpassing the GT-depth version. Compared to F³Loc mono, our method achieves a 52.9-point SR gain. Importantly, UnLoc maintains an SR above 80% even on 15-frame sequences, whereas F³Loc (with either the original encoder or DepthAnything) requires at least 50 frames to reach this level. Excelling on short sequences is particularly relevant for real-world applications, where time-to-localize is critical.

For completeness, we also include LASER (Min et al., 2022) results (as reported in Chen et al. (2024a)), which lag behind both F³Loc and UnLoc by a wide margin.

**Results on LaMAR.** Table 2 reports results on the LaMAR HGE dataset. Since F³Loc does not provide a pre-trained model for this dataset, we evaluate it using a model trained by us. LaMAR is a challenging real-world dataset with both small environments (offices and rooms) and large ones (halls and long corridors). The monodepth predictor of F³Loc struggles in these conditions, yield-

| Model | SR@1m (%; T=100) ↑ | (T=50) ↑ | (T=35) ↑ | (T=20) ↑ | (T=15) ↑ |
|---|---|---|---|---|---|
| GT Depth | 100.0 | 100.0 | 57.1 | 53.8 | 22.2 |
| F$^3$Loc mono | 0.0 | 0.0 | 0.0 | 0.0 | 0.0 |
| **UnLoc** w/o post-processing | **50.0** | 20.0 | 0.0 | 15.4 | 0.0 |
| **UnLoc** | **50.0** | **40.0** | **57.1** | **30.8** | **16.7** |

Table 4: **Localization** on the LaMAR CAB dataset with models trained on LaMAR HGE. We report the success rates (SR) for sequence lengths 100, 50, 35, 20, and 15.

ing very low success rates even for long sequences. Replacing the encoder of F$^3$Loc with Depth Anything v2 already yields more accurate results than the original model.

The proposed UnLoc, equipped with uncertainty modeling and uncertainty-weighted post-processing, brings substantial gains. Compared to F$^3$Loc, the success rate rises from 1.2% to 50.6% (a 42.2-fold improvement) on 15-frame sequences and from 36.4% to 100% (a 2.7-fold improvement) on 100-frame sequences. On the cropped dataset variant (Chen et al., 2024a), success rates increase by 25.0 to 80.8 percentage points, depending on the sequence length. Both uncertainty modeling and post-processing contribute significantly to these improvements. These results demonstrate that UnLoc markedly improves upon the state of the art in complex, realistic scenarios by leveraging off-the-shelf depth predictors with uncertainty estimation. The improvements are complementary and each component contributes to the overall accuracy.

Figure 3 shows examples of floorplan depth predictions and uncertainties. UnLoc consistently outperforms F$^3$Loc on these challenging cases. Notably, in the last example, our method returns the correct depth while the ground truth is incorrect.

Table 4 presents results on the LaMAR CAB building using models trained on the LaMAR HGE dataset. While this is not a strict zero-shot scenario – since the same sensor is used in a different building – it effectively demonstrates the generalization capabilities of UnLoc. Our approach achieves accurate localization in this new environment, whereas F$^3$Loc fails completely. We provide additional results on this cross-domain task in the supp. mat (see Sec. A.2.2).

**Runtime.** Table 3 presents the average runtime per frame for each method. As expected, using an off-the-shelf depth prediction network incurs a higher computational cost than the custom network used in Chen et al. (2024a). On the Gibson dataset, all methods run efficiently, though our proposed method is slightly slower due to the more complex depth predictor. On the LaMAR dataset, all methods require approximately one second per frame. In practical applications, a lower frame rate is acceptable since new frames may not always provide significantly new information at high rates. Also, UnLoc excels on short sequences, achieving accurate localization with fewer frames. The post-processing takes, on average, 0.96 seconds once at the end of the process.

**Summary.** The results on both synthetic and real-world datasets confirm that UnLoc outperforms the state of the art in terms of accuracy and robustness. We achieve significant improvements in sequential visual floorplan localization by addressing the limitations of the state of the art and leveraging uncertainty-aware depth predictions from pre-trained models. UnLoc maintains real-time performance while enhancing scalability and generalization to new scenes.

## 4.1 ABLATION STUDIES

**Monocular Depth Networks.** We further examine the impact of different encoder networks on our method's performance using the LaMAR HGE dataset (Table 5). Specifically, we evaluate the general-purpose encoder DINOv2 (Oquab et al., 2024), the monodepth encoder DepthPro (Bochkovskii et al., 2024), and three encoder variants of Depth Anything v2 (Yang et al., 2024b) – large (L), base (B), and small (S). The results show that the monodepth encoders DepthPro and Depth Anything v2 (L) outperform the similarly-sized general-purpose encoder DINOv2, with Depth Anything v2 (L) achieving the highest overall performance. This indicates that pre-trained monodepth encoders provide more effective features for floorplan depth prediction. Within the Depth Anything v2 models, performance scales positively with model size. Importantly, incorporating the proposed depth uncertainty estimation consistently improves success rates across *all* encoders. For instance, incorporating depth uncertainties with the base model elevates its performance to match that of the

| Model | T=100 | T=50 | T=35 | T=20 | T=15 |
|---|---|---|---|---|---|
| DINOv2 (L) | 90.9 | 45.8 | 20.0 | 9.5 | 3.5 |
| DINOv2 (L) w/ Uncertainty | **100.0** | **54.2** | **31.4** | **15.9** | **5.9** |
| DepthPro | 90.9 | 62.5 | 40.0 | 17.5 | 4.7 |
| DepthPro w/ Uncertainty | **100.0** | **70.8** | **51.4** | **31.7** | **14.1** |
| Depth Anything V2 (L) | **100.0** | 66.7 | 42.9 | 23.8 | 9.4 |
| Depth Anything V2 (L) w/ Uncertainty | **100.0** | **75.0** | **60.0** | **36.5** | **20.0** |
| Depth Anything V2 (B) | 90.9 | 62.5 | 28.6 | 12.7 | 2.4 |
| Depth Anything V2 (B) w/ Uncertainty | **100.0** | **66.7** | **42.9** | **30.2** | **12.9** |
| Depth Anything V2 (S) | 81.8 | 41.7 | 22.9 | 7.9 | 4.7 |
| Depth Anything V2 (S) w/ Uncertainty | **100.0** | **54.2** | **37.1** | **17.5** | **5.9** |

Table 5: **Sequential localization with UnLoc using different encoders** on LaMAR HGE (Sarlin et al., 2022). Results are without post-processing. Success rate for different sequence lengths (localization is a success if the last 10 frames are within 1m of GT). We compare DINOv2 (Oquab et al., 2024) with DepthPro (Bochkovskii et al., 2024) and different pre-trained models of Depth Anything V2 (Yang et al., 2024b) (L: large, B: base, S: small), both with and without uncertainty estimation.

large variant. This demonstrates that uncertainty estimation effectively compensates for smaller model sizes, enhancing robustness without incurring additional computational costs.

**Efficiency Analysis.** Fig. 4 illustrates the SR on the Gibson(t) dataset versus model size (left) and runtime (right). To show a fair model comparison, we do not perform post-processing optimization here. Our method improves substantially over $F^3$Loc even with smaller models: Using the small Depth Anything v2 model, it achieves about a 10 percentage point higher success rate compared to $F^3$Loc fusion, while maintaining similar runtime. This highlights the efficiency of our approach in terms of accuracy versus computational resources.

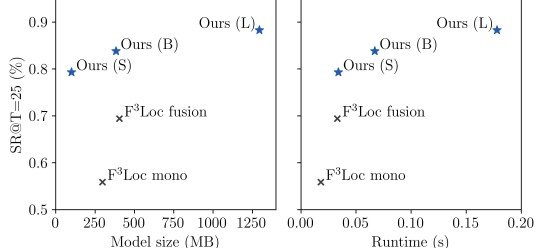

Figure 4: **Efficiency Analysis.** Performance of sequential localization versus model size (left) and runtime (right) on an NVIDIA Quadro RTX 6000 GPU. The success rate (SR) is defined as the percentage of sequences of length $T = 25$ for which the posterior remains within an error radius of 1m in the 10 frames. The values are averaged over all test sequences from the Gibson(t) dataset.

## 5 CONCLUSION

We propose a visual floorplan localization method, UnLoc, that addresses key limitations of prior approaches by incorporating uncertainty estimation into depth prediction and leveraging pre-trained monodepth models. Modeling depth predictions as probability distributions allows us to quantify uncertainty, leading to improved accuracy, especially in challenging environments. Experiments on both synthetic and real-world datasets demonstrate that our method significantly outperforms state-of-the-art approaches in terms of accuracy and robustness while operating efficiently. Our approach offers a practical and scalable solution for visual floorplan localization across diverse indoor environments.

REPRODUCIBILITY STATEMENT

We have taken several steps to ensure reproducibility. The code required for data preprocessing, model training, model evaluation, along with trained models, is publicly available at https://github.com/matthias-wueest/UnLoc. Training procedures are described in A.1.1 and dataset creation details are provided in A.1.2.

**Acknowledgments.** Francis Engelmann acknowledges support from an SNSF PostDoc mobility fellowship. This research was also supported in part by an academic gift from NVIDIA and Meta. The authors gratefully acknowledge this support.

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

## A  APPENDIX

This section provides further implementation details (Sec. A.1), as well as additional quantitative (Sec. A.2) and qualitative results (Sec. A.3).

### A.1  ADDITIONAL IMPLEMENTATION DETAILS

#### A.1.1  TRAINING DETAILS

For our evaluations, we train the models on the full training split of the respective dataset for 50 epochs. The training set is shuffled at the beginning of each epoch to ensure variability, and the models are trained using a batch size of four. We employ Adam (Kingma, 2014) with a fixed learning rate of $1 \times 10^{-3}$.

#### A.1.2  DATASET DETAILS: LaMAR HGE AND CAB

To tailor the LaMAR dataset (Sarlin et al., 2022) for our experimental requirements, we apply a series of preprocessing steps and customizations, following a similar strategy to Chen et al. (2024a). The dataset originally includes images captured by iOS and Hololens devices. We restrict ourselves to the images recorded by the iOS devices from the mapping part of the dataset. We filter the dataset to retain only the indoor poses, excluding outdoor data. For the *LaMAR HGE Complete* dataset, we utilize all indoor camera poses recorded at the HGE location. In contrast, the *LaMAR HGE Cropped* dataset focuses on a smaller region, specifically a reduced area around the center of the

HGE floorplan. To avoid transitions between different floorplans in the *LaMAR CAB* dataset, we restrict the data to a single floor, specifically floor F of the CNB building.

We use the official building floorplans as basis and apply selected manual modifications. Room numbers, stairs, and doors connecting two corridors that are typically open are removed, while all other doors are represented as solid walls. For the mapping of LaMAR 6 DoF poses to 2D floorplans we proceed as follows: We identify four significant building entrance doorways visible in the image sequences, positioned to span the entire floorplan area. For each doorway, we determine: (1) the LaMAR ground truth (x, y) position in meters when the camera crosses the threshold, and (2) the corresponding pixel location of that doorway in the floorplan. The pixel positions are manually marked by visually identifying doorway locations in the architectural floorplan and examining the images to determine when the camera crosses each threshold. We then fit a 2D affine transformation (6 parameters, 8 constraints from 4 point pairs) mapping meter coordinates to pixel coordinates using least squares. This transformation is applied to all ground truth positions. Yaw angles are extracted directly from the ground truth quaternions. While we lack independent ground truth for the 2D poses, multiple consistency checks suggest the alignment is reliable: low affine fit residuals, trajectories align with building geometry, and raycast depths match observed structure. Using these 2D poses, the ground truth floorplan depths required for training are derived from the modified floorplans via the raycasting method from Chen et al. (2024a).

The splits are fixed prior to experimentation and are not adjusted in any way to favor any method. For LaMAR HGE, the split includes 12 sessions used for training, one for validation, and three for testing. The test sessions were selected with the following goals: First, to include the session illustrated in Figure 11 of $F^3$Loc Chen et al. (2024a) for qualitative comparison, and second, to ensure diversity across lighting and occlusion conditions. The three test sessions therefore span (i) daytime with moderate occlusion, (ii) daytime with heavy occlusion due to an ongoing exhibition, and (iii) nighttime with minimal occlusion. For LaMAR CAB, the split includes 1 session for training (only in case of fine-tuning) and 2 for testing. Here, all sessions had similar daytime and occlusion.

## A.2 Additional quantitative results

### A.2.1 Single-Frame Localization

In this section, we evaluate the performance of each method in single-frame localization tasks. While our primary focus is sequential localization, analyzing single-frame performance provides valuable insights into the individual components of our approach.

Table 6 presents the results on the Gibson(t) (Xia et al., 2018), Structured3D (Zheng et al., 2020), and LaMAR HGE (Sarlin et al., 2022) datasets. We report the localization recall, i.e., the percentage of frames localized within specified accuracy thresholds. Across all datasets, making use of an off-the-shelf monodepth encoder instead of a custom depth predictor leads to substantial improvements in all metrics. Specifically, $F^3$Loc with Depth Anything v2 outperforms the original $F^3$Loc by a significant margin in terms of both mean absolute error (MAE; in meters) and localization recall. This demonstrates the effectiveness of leveraging pre-trained depth models.

As anticipated, incorporating depth uncertainties into the single-frame estimation does not have a large impact on the Gibson(t) and Structured3D datasets. This can be attributed to the fact that uncertainty estimation is particularly beneficial when combining multiple measurements over time, as in sequential localization. However, on the challenging LaMAR dataset, which features complex environments with varying occlusions and narrow fields of view, uncertainty estimation leads to improvements in recall across most thresholds. This suggests that uncertainty modeling helps with handling complex and real-world scenarios.

### A.2.2 Generalization between datasets

To demonstrate the generalization capabilities of our proposed UnLoc, we train models on the LaMAR HGE dataset and assess their performance on LaMAR CAB. The evaluation includes a comparison between UnLoc and $F^3$Loc mono across various scenarios, with and without fine-tuning on a limited number of frames (n=200) from the LaMAR CAB dataset. The results, presented in Table 7 and 8, demonstrate that UnLoc without any additional training on the target domain converges

within 2m of the ground truth (GT) for trajectories of length T=100, outperforming the $F^3$Loc mono variants. Furthermore, the results show that fine-tuning on a small amount of target-domain data leads to performance increases in both the sequential and single-image localization. This indicates a certain adjustment to new datasets is still necessary for optimal results.

### A.2.3 REAL-TIME CONSIDERATIONS FOR LARGE-SCALE FLOORPLANS

As shown in Table 3 of the main paper, matching operations dominate the runtime for large floor-plans. To further reduce the computational cost on large-scale floorplans, we investigate two strate-gies: (1) decreasing the frequency of observation updates, performing them only after every $\Delta T$ transition updates, and (2) employing a coarser grid for the matching with the floorplan.

Table 9 presents the sequential localization performance on the LaMAR HGE dataset with less frequent observation updates. Evaluating our proposed UnLoc on trajectories of length T=100, halving the observation frequency has no negative impact on localization accuracy. Even when observation updates are performed at every 10th step, UnLoc achieves convergence in nearly every second case. Results are shown without post-processing optimization.

An ablation study on grid resolution is provided in Table 10. The uncertainty-aware matching stage of the UnLoc pipeline compares probabilistic floorplan depth predictions with occupancy infor-mation from the floorplan. The computational cost and mapping quality in this stage are directly influenced by the grid resolution. A finer grid enhances mapping quality but leads to increased com-putational cost. In our experiments in the main paper, we assume a grid resolution of 0.1m $\times$ 0.1m. The results show that coarsening the grid to 0.25m $\times$ 0.25m reduces the matching time by a factor of over six, at the expense of only a slight decrease in localization performance.

### A.2.4 DEPTH UNCERTAINTY DISTRIBUTION

We compare two approaches for modeling depth uncertainty in UnLoc: a Laplacian model (ours) and a Gaussian model, as proposed by Roessle et al. (2022). In the Gaussian model, the obser-vation likelihood in Eq. 1 is modified to a product of independent Gaussian distributions instead of Laplacian distributions, with $\tilde{d}_{t,j}$ as mean parameter and $\tilde{b}_{t,j}$ as standard deviation parameter. Correspondingly, the loss function in Eq. 5 is adapted to reflect the negative log-likelihood of the Gaussian distribution rather than the Laplacian distribution.

Table 11 reports the sequential localization performance on the LaMAR HGE dataset for vary-ing sequence lengths $T$. Although both models achieve a perfect success rate for long sequences ($T = 100$), our Laplacian model consistently outperforms the Gaussian model on shorter sequences, demonstrating superior convergence behavior.

## A.3 ADDITIONAL QUALITATIVE RESULTS

### A.3.1 OBSERVATION LIKELIHOOD

A qualitative comparison of the predicted observation likelihoods between $F^3$Loc and UnLoc on the Gibson(t) dataset is presented in Figure 5. As shown, our UnLoc approach consistently delivers more accurate predictions than $F^3$Loc. Moreover, UnLoc with uncertainty takes advantage of predicted low uncertainties in floorplan depth estimates, enabling it to more effectively reduce the observation likelihood for less probable states.

### A.3.2 POSTERIOR PROBABILITY

Figure 6 compares the posterior probability evolution of UnLoc with those of $F^3$Loc mono and $F^3$Loc fusion across three trajectories from the Gibson(t) dataset. In all three cases, UnLoc outper-forms both baselines, achieving substantially faster convergence to the true pose. Notably, UnLoc converges within five or fewer frames in each trajectory, a result that generally holds for around two thirds of the trajectories of the Gibson(t) dataset (without post-processing ), as shown in Table 1 of the main paper.

| | Model | Depth pred. | | Recall (%) ↑ | | | | | | |
|---|---|---|---|---|---|---|---|---|---|---|
| | | MAE (m) ↓ | Cos Sim ↑ | 0.1m | 0.5m | 1m | 1m 30° | 2m | 5m | 10m |
| Gibson(t) | GT Depth | 0.00 | 1.000 | 47.1 | 79.0 | 80.3 | 79.8 | 82.4 | 90.6 | 98.5 |
| | F³Loc fusion | 0.28 | 0.980 | 15.6 | 53.6 | 59.1 | 57.6 | 64.4 | 82.7 | 97.3 |
| | F³Loc mono | 0.43 | 0.976 | 7.3 | 40.3 | 49.1 | 47.3 | 55.5 | 79.6 | 97.7 |
| | F³Loc mono* | 0.42 | 0.973 | 8.2 | 39.2 | 46.5 | 44.6 | 54.5 | 79.7 | 97.7 |
| | + Depth Anything v2 | **0.23** | **0.981** | 19.3 | **62.5** | **66.5** | **65.6** | **70.6** | **85.2** | **98.1** |
| | **UnLoc** | 0.25 | **0.981** | **19.7** | 61.1 | 64.7 | 63.8 | 68.7 | 84.4 | 97.6 |
| Structured3D | GT Depth | 0.00 | 1.000 | 32.2 | 63.4 | 65.0 | 64.3 | 68.6 | 81.6 | 96.4 |
| | PF-net (copied from Chen et al. (2024a)) | – | – | 0.2 | 1.3 | 3.2 | 0.9 | – | – | – |
| | LASER (copied from Chen et al. (2024a)) | – | – | 0.7 | 6.4 | 10.4 | 8.7 | – | – | – |
| | F³Loc mono (copied from Chen et al. (2024a)) | – | – | 1.5 | 14.6 | 22.4 | 21.3 | – | – | – |
| | F³Loc mono* | 0.51 | 0.971 | 1.7 | 19.2 | 27.7 | 26.5 | 35.1 | 62.6 | 94.3 |
| | + Depth Anything v2 | **0.37** | **0.979** | **5.5** | **34.2** | **40.4** | **39.3** | **45.8** | 68.0 | 94.9 |
| | **UnLoc** | 0.39 | **0.979** | 5.3 | 33.9 | 38.8 | 37.6 | 44.9 | **68.3** | **95.1** |
| LaMAR | GT Depth | 0.00 | 1.000 | 14.3 | 44.7 | 49.2 | 49.2 | 49.5 | 52.5 | 56.3 |
| | F³Loc mono* | 3.29 | 0.930 | 0.0 | 0.1 | 0.8 | 0.8 | 1.2 | 3.4 | 8.6 |
| | + Depth Anything v2 | 2.08 | 0.957 | **0.5** | 6.8 | 11.3 | 11.3 | 13.9 | 17.6 | 22.8 |
| | **UnLoc** | **2.02** | **0.958** | 0.4 | **11.3** | **17.1** | **17.1** | **19.2** | **22.9** | **27.5** |

Table 6: **Single-frame localization** on the Gibson(t) (Xia et al., 2018), LaMAR HGE (Sarlin et al., 2022), and Structured3D (Zheng et al., 2020) datasets. We report the mean absolute error (MAE) in meters and the cosine similarity (Cos Sim) of the depth feature embeddings to assess the accuracy of the depth prediction. For camera localization accuracy, we report the recall (%) at various distance thresholds. The first row in each dataset section shows the localization performance using ground truth (GT) depth, providing an upper bound. * indicates that the F³Loc model was trained by us.

| | T=100, N=2 | | | T=50, N=5 | T=35, N=7 | T=20, N=13 | T=15, N=18 |
|---|---|---|---|---|---|---|---|
| Model | SR@2m (%) ↑ | RMSE (succ.) ↓ | RMSE (all) ↓ | SR@2m (%) ↑ | @2m (%) ↑ | @2m (%) ↑ | @2m (%) ↑ |
| GT Depth | 100.0 | 0.09 | 0.09 | 100.0 | 57.1 | 53.8 | 22.2 |
| F3Loc mono* | 0.0 | nan | 14.90 | 0.0 | 0.0 | 0.0 | 0.0 |
| F3Loc mono* (ft: 5 ep.) | 0.0 | nan | 6.86 | 20.0 | 0.0 | 0.0 | 0.0 |
| F3Loc mono* (ft: 10 ep)) | 50.0 | 0.66 | 3.59 | 0.0 | 0.0 | 0.0 | 0.0 |
| **UnLoc** | **100.0** | 0.71 | 0.71 | 40.0 | 57.1 | 30.8 | 16.7 |
| **UnLoc** (ft: 5 ep.) | **100.0** | 0.42 | 0.42 | **100.0** | **85.7** | 30.8 | **33.3** |
| **UnLoc** (ft: 10 ep.) | **100.0** | **0.35** | **0.35** | **100.0** | 71.4 | **38.5** | 22.2 |

Table 7: **Sequential localization** on the LaMAR CAB dataset (Sarlin et al., 2022) with models trained on LaMAR HGE. We report the success rates (SR) and the RMSE over the successful and all sequences when the sequence length (T) is 100. We report the success rate for all other lengths, considering a localization a success if the accuracy of the last 10 frames is within 2m of the ground truth (GT). We also show the number of sequences tested (N) in each setting. In the first row, we report the localization accuracy with the GT depth. "ft" indicates the model was fine-tuned on the LaMAR CAB dataset (Sarlin et al., 2022) for the specified number of epochs (ep.). * indicates the F³Loc model was trained by us.

| | Model | Depth pred. | | Recall (%) ↑ | | | | | | |
|---|---|---|---|---|---|---|---|---|---|---|
| | | MAE (m) ↓ | Cos Sim ↑ | 0.1m | 0.5m | 1m | 1m 30° | 2m | 5m | 10m |
| | GT Depth | 0.00 | 1.000 | 17.5 | 32.5 | 33.0 | 33.0 | 36.0 | 42.0 | 50.5 |
| | F3Loc mono* | 2.79 | 0.892 | 0.0 | 0.0 | 0.5 | 0.0 | 1.0 | 3.5 | 12.5 |
| | F3Loc mono* (ft: 5 ep.) | 1.47 | 0.940 | 0.0 | 0.5 | 1.0 | 0.5 | 3.5 | 7.5 | 20.0 |
| | F3Loc mono* (ft: 10 ep.) | 1.52 | 0.940 | 0.0 | 0.5 | 1.0 | 1.0 | 2.5 | 7.5 | 19.0 |
| | **UnLoc** | 1.39 | 0.970 | 0.0 | 1.0 | 2.5 | 2.5 | 3.5 | 9.0 | 24.0 |
| | **UnLoc** (ft: 5 ep.) | **0.78** | **0.974** | 0.0 | **4.0** | **9.0** | **8.0** | **11.0** | 18.0 | **30.0** |
| | **UnLoc** (ft: 10 ep.) | 0.79 | 0.971 | 0.0 | 3.0 | 5.5 | 5.5 | 10.0 | **20.0** | **30.0** |

Table 8: **Single-frame localization** on the LaMAR CAB dataset (Sarlin et al., 2022) with models trained on LaMAR HGE. We report the mean absolute error (MAE) in meters and the cosine similarity (Cos Sim) to assess the accuracy of the depth prediction. For camera localization accuracy, we report the recall (%) at various distance thresholds. The first row shows the localization performance using ground truth (GT) depth, providing an upper bound. "ft" indicates the model was fine-tuned on LaMAR CAB for the specified number of epochs (ep.). * indicates that the F³Loc model was trained by us.

| | SR@1m (%) ↑ | | | |
|---|---|---|---|---|
| Model | $\Delta$t=1 | $\Delta$t=2 | $\Delta$t=4 | $\Delta$t=10 |
| GT Depth | 100.0 | 100.0 | 100.0 | 45.5 |
| F3Loc mono* | 36.4 | 27.3 | 18.2 | 0.0 |
| **UnLoc** w/o post-processing | **100.0** | **100.0** | **72.7** | **45.5** |

Table 9: **Sequential Localization** on LaMAR HGE dataset (Sarlin et al., 2022) with less frequent observation updates. We report the success rates (SR) for a sequence length (T) of 100 and different interval lengths between observations ($\Delta$t). We consider a localization a success if the accuracy of the last 10 frames is within 1m of the ground truth (GT). In the first row, we report the localization accuracy with the GT depth. * indicates the F$^3$Loc model was trained by us.

| | SR@1m (%) ↑ | | | | | Timing (s) | |
|---|---|---|---|---|---|---|---|
| Grid resolution | T=100 | T=50 | T=35 | T=20 | T=15 | Feat. extr. | Matching |
| 0.1m x 0.1m | **100.0** | **75.0** | **60.0** | **36.5** | **20.0** | **0.185** | 0.880 |
| 0.25m x 0.25m | 90.9 | 70.8 | 57.1 | 30.2 | 15.3 | **0.185** | **0.140** |

Table 10: **Sequential localization and timing** of our UnLoc approach without post-processing on the LaMAR HGE dataset (Sarlin et al., 2022) for different grid resolutions. We report the success rates (SR) various sequence lengths (T). We consider a localization a success if the accuracy of the last 10 frames is within 1m of the ground truth (GT). We report the runtimes (feature extraction and matching) averaged over all test sequences on an NVIDIA Quadro RTX 6000 GPU.

| Model | SR@1m (%; T=100) ↑ | (T=50) ↑ | (T=35) ↑ | (T=20) ↑ | (T=15) ↑ |
|---|---|---|---|---|---|
| **UnLoc** w/ Laplacian | **100.0** | **54.2** | **37.1** | **17.5** | **4.7** |
| **UnLoc** w/ Gaussian | **100.0** | 41.7 | 20.0 | 6.3 | 0.0 |

Table 11: **Sequential Localization with UnLoc using different uncertainty models.** Comparison of depth uncertainty distributions on LaMAR HGE (Sarlin et al., 2022) without post-processing using a Depth Anything v2 (Small) depth network. Success rates (SR) are reported for varying sequence lengths $T$.

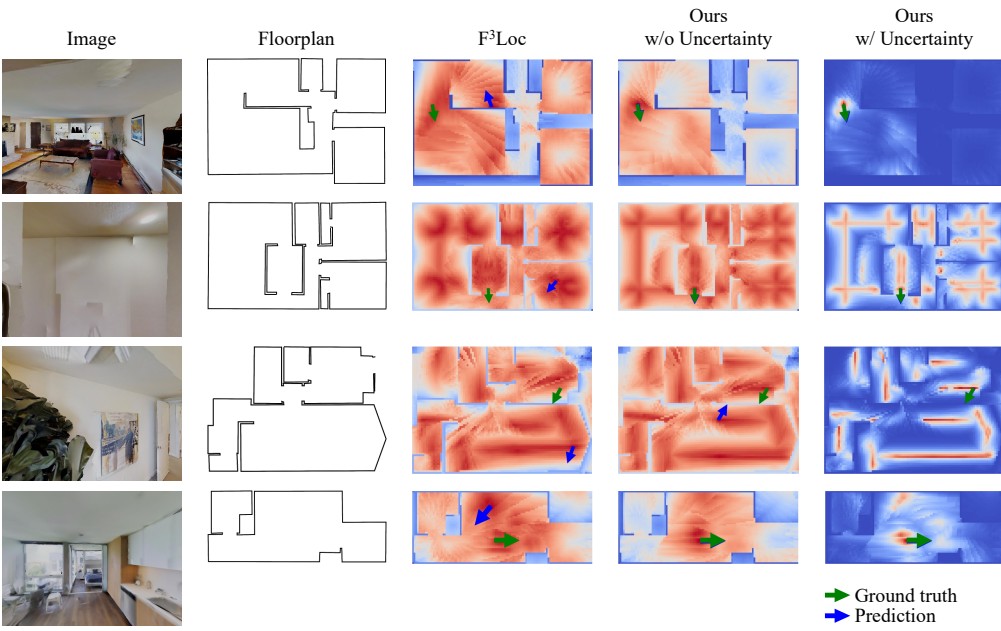

Figure 5: **Observation likelihoods.**

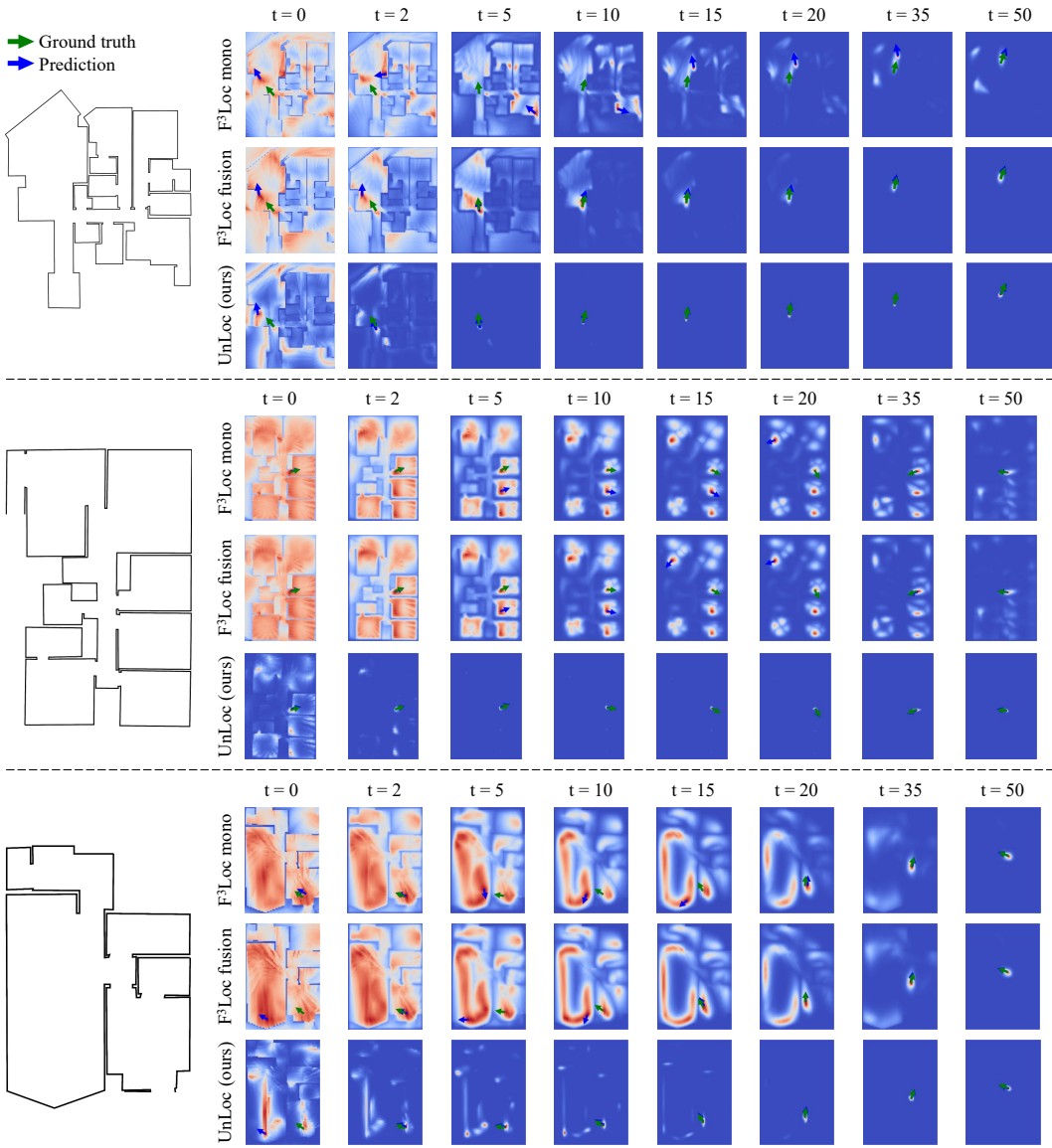

Figure 6: **Posterior evolution.**

