# OpenReview forum: "UnLoc: Leveraging Depth Uncertainties for Floorplan Localization"
_ICLR.cc/2026/Conference — ICLR 2026 Poster_

### Official Review · Reviewer_PYiR · 2025-10-19

**Soundness:** 3
**Presentation:** 3
**Contribution:** 1
**Rating:** 2
**Confidence:** 4

**Summary:**

This paper proposes combining pre-trained depth models with uncertainty modeling to improve the accuracy of floor-plan localization. Experiments are conducted on multiple datasets, including both real-world and synthetic scenes.

**Strengths:**

1. The paper is generally well written and easy to follow.

2. It makes sense to leverage uncertainty information for localization.

3. The proposed method achieves state-of-the-art performance on two floor-plan localization benchmarks.

**Weaknesses:**

1. The paper mainly integrates existing methods and applies them to a relatively narrow task—floor-plan localization. While the topic is relevant, the contribution appears incremental rather than conceptually novel. This paper does not introduce any new contribution in terms of problem formulation; it focuses on a narrow task with clearly defined inputs and outputs.

2. As shown in Table 6, F3Loc + Depth Anything V2 already achieves higher accuracy than UnLoc on several datasets. This suggests that much of the performance gain may stem from the use of off-the-shelf pre-trained depth foundation models rather than from the new proposed method in this paper. The paper, therefore, reads more as an application of pre-trained depth models to a specific downstream task than as a novel algorithmic/theory contribution.

3. The discussion on uncertainty is underdeveloped. Uncertainty in neural network predictions can generally be categorized into aleatoric and epistemic uncertainty [1], yet this paper lacks a theoretical explanation or analysis of which type is modeled or how it contributes to performance improvements.

4. The data preprocessing and benchmark setup are not sufficiently transparent. The paper appears to introduce newly processed data and custom split strategies from public datasets, which raises concerns about reproducibility. It is unclear whether the test set division used in this work aligns with the official benchmark split and whether the authors may have chosen a split that yields the best performance. Providing clear details on dataset handling and evaluation would significantly improve the credibility of the results. This paper does not provide the recommended Reproducibility statement and the Use of Large Language Models (LLMs) statement.


[1] Alex Kendall et al., What uncertainties do we need in Bayesian deep learning for computer vision?, NeurIPS 2017.

**Questions:**

I am particularly curious about how this work is evaluated on the LaMAR dataset. To my knowledge, LaMAR does not include publicly available floor plans. Could the authors clarify where the floor plan data used in this paper originated? The description in the Appendix is quite brief and would benefit from a more detailed explanation.

Moreover, LaMAR does not provide ground-truth poses for the test set, and the paper does not specify the session divisions used. Since the evaluation is based on estimating 2D poses on floor plans, it remains unclear how these poses were submitted or compared within the LaMAR benchmark to produce the results shown in Table 2. Additionally, even if the official floor plans were obtained, how were the 6Dof camera poses from LaMAR aligned with the 2D pose in the floor plans? If you use custom's align algorithm to calculate the ground-truth label of 2D poses, how can you ensure that this ground-truth itself is reliable?

A detailed explanation of the data processing pipeline and evaluation protocol would be very helpful.

I also noted that F3Loc did not include such evaluation details—it only released the link to the Gibson Floorplan Localization Dataset. This raises further concerns about the fairness and transparency of comparing F3Loc and the proposed method on the LaMAR HGE/CAB benchmark. Overall, this area still lacks openly available datasets and standardized baselines, making reproducibility a critical issue at this stage.

If the authors could provide a more comprehensive description of the evaluation methodology and experimental setup on LaMAR, I would consider improving my overall rating. Furthermore, I strongly encourage the authors to release the processed version of the LaMAR data used in this work. Since the paper involves substantial preprocessing of the raw LaMAR dataset, referring to it as “widely used” without sharing those processed resources seems somewhat inappropriate and limits the community’s ability to validate the claimed improvements in accuracy and reproducibility.

---

> ### Author Response · Authors · 2025-11-22
>
> We thank the reviewer for the thorough evaluation and for noting that our paper is “well written and easy to follow”, that “it makes sense to leverage uncertainty information”, and that we “achieve state-of-the-art performance”. We address the concerns below and commit to significantly enhancing transparency (see separate comment below for this).
>
> ### Integrates existing methods
> We would like to clarify that our contributions represent methodological innovation, not mere application:
>
> (1) Principled uncertainty-aware formulation: To our knowledge, this is the first work to explicitly model, predict, and propagate geometric observation uncertainty in a sequential Bayesian framework for floorplan localization.
>
> (2) Empirical proof of necessity: Ablations show each element contributes meaningfully, and together they yield substantial gains, demonstrating the method is more than an incremental integration.
>
> (3) Design choices with broader relevance: Our framework includes components of general value, such as the Laplace distribution for robust geometric matching, uncertainty-weighted fusion, uncertainty-weighted refinement, and a dual-head architecture trained on top of a frozen encoder.
>
> ### Narrow task formulation
>
> We focus on sequential floorplan localization because it allows rigorous analysis and is central to robotics and AR. While the task is specific, the methodological insights generalize to other geometry-aware problems: uncertainty-aware fusion, frozen‐encoders with learned uncertainty heads, robust treatment of outlier regions and representation transfer in geometric pipelines. These design patterns are applicable to SLAM, tracking, visual odometry, and other sequential geometric reasoning tasks.
>
> ### “F3Loc + Depth Anything V2 already achieves higher accuracy”
>
> Please note that the referred Table 6 reports only single-frame localization. Our main contribution is sequential localization (Tables 1-2). Averaged over all sequence lengths, our method improves SR@1m over “F3Loc+DA” by +15.0% (Gibson), +24.1% (LaMAR HGE Original), and +26.0 (LaMAR HGE Cropped), with even larger gains (24.9% to 46.8%) for shorter sequences (T=15, 20). This shows our method is not a simple application of pre-trained depth models but a principled framework that substantially improves sequential localization.
>
> ### Uncertainty discussion
>
> We appreciate this feedback and have significantly expanded Sections 2 and 3.5. Our uncertainty is primarily aleatoric, capturing ambiguity from glass walls, doorways, and occlusions. We revised Section 2 to clarify this and added the suggested reference. Section 3.5 now better explains why Laplace likelihoods were selected for this task.
>
> ### “Widely used” statement
> We removed this wording, as it does not accurately describe the processed dataset.
>
> ### Reproducibility statement
> We have added a reproducibility statement to the paper.
>
> ### LLM statement
>
> Following the ICLR author guidelines, an LLM statement is required only if LLMs contributed substantially to research ideation or content. In our case, LLMs were used only to aid or polish writing (see OpenReview submission form).

---

> ### Author Response · Authors · 2025-11-22
>
> ### Train/test splits
> Structured3D and Gibson follow official splits.
> For LaMAR, we contacted the F3Loc team but the splits could not be shared, so we created our own splits using only the iOS sessions of the dataset.
>
> The test sessions were chosen to (a) include the session illustrated in Figure 11 of F3Loc (ios_2022-07-03_16.00.37_000) for qualitative comparison, and (b) to ensure diversity across lighting and occlusion conditions. The splits were fixed prior to experimentation and were not adjusted in any way to favor any method. These details are now also in Section (A.1.2).
>
> #### LaMAR HGE:
> Train:
> - ios_2022-01-12_15.15.53_000
> - ios_2022-01-18_17.05.03_000
> - ios_2022-01-18_17.10.39_000
> - ios_2022-01-20_16.52.33_001
> - ios_2022-01-25_14.34.24_002
> - ios_2022-01-25_14.57.49_000
> - ios_2022-01-25_15.13.54_000
> - ios_2022-06-13_10.45.07_000
> - ios_2022-06-13_15.59.36_000
> - ios_2022-06-14_17.12.28_000
> - ios_2022-06-30_15.55.53_000
> - ios_2022-07-01_15.18.09_000
>
> Validation:
> - ios_2022-01-25_15.13.54_002
>
> Test:
> - ios_2022-07-01_15.45.08_000
> - ios_2022-01-18_17.10.39_000
> - ios_2022-07-03_16.00.37_000
>
> #### LaMAR CAB
>
> Train (for fine-tuning only):
> - ios_2022-06-22_13.33.34_002
>
> Test:
> - ios_2022-06-02_14.48.13_000
> - ios_2022-06-20_18.27.53_000
>
>
>
> ### Origin of floorplan data
> Following F3Loc, we obtained official floorplans from ETH Zurich. Redistribution is restricted, which also explains why F3Loc did not release their processed dataset (see [Link](https://ethz.ch/staffnet/en/service/rooms-and-buildings/building-orientation/gebaeude.html?args0=HG)). Despite this limitation, LaMAR represents one of the few large-scale, real-world indoor benchmarks with accurate ground truth poses.
>
> ### Pose mapping:
>
> We added the following details to the Appendix A.1.2.:
>
> “For the mapping of LaMAR 6 DoF poses to 2D floorplans we proceed as follows: We identify four significant building entrance doorways visible in the image sequences, positioned to span the entire floorplan area. For each doorway, we determine: (1) the LaMAR ground truth (x, y) position in meters when the camera crosses the threshold, and (2) the corresponding pixel location of that doorway in the floorplan. The pixel positions are marked by visually identifying doorway locations in the architectural floorplan and examining the images to determine when the camera crosses each threshold. We then fit a 2D affine transformation (6 parameters, 8 constraints from 4 point pairs) mapping meter coordinates to pixel coordinates using least squares. This transformation is applied to all ground truth positions. Yaw angles are extracted directly from the ground truth quaternions. While we lack independent ground truth for the 2D poses, multiple consistency checks suggest the alignment is reliable: low affine fit residuals, trajectories align with building geometry, and raycast depths match observed structure.”
>
> ### What we will release
>
> Due to floorplan restrictions, we commit to releasing upon acceptance: floorplan preprocessing documentation (floorplan modifications detailed in Appendix A.1.2), data pre-processing incl. pose mapping code, session IDs/splits, complete training/evaluation code, trained models.
>
> While floorplan distribution is restricted, researchers with floorplan access can reproduce our evaluation or apply the methodology to their own datasets.
>
>
>
> ### Final remark
> We hope these detailed clarifications and our commitment to comprehensive documentation address the reviewer's concerns about transparency and reproducibility.

---

### Official Review · Reviewer_Lj7y · 2025-10-24

**Soundness:** 3
**Presentation:** 3
**Contribution:** 2
**Rating:** 4
**Confidence:** 3

**Summary:**

This paper presents UnLoc (Uncertainty-aware Localization), a method for sequential camera localization within 2D floorplans. The input of the method is:
1/ a sequence of t RGB images,
2/ relative poses between these images,
3/ gravity direction
4/ camera intrinsics, and
5/ geometry layout of the floorplan

It aims to address two key shortcomings of previous state-of-the-art methods: the lack of uncertainty modeling in depth predictions and the need for custom, environment-specific depth networks for each environment (e.g. in $F^3Loc$). The paper reports strong performance compared to previous methods on LaMAR HGE dataset.

**Strengths:**

(+) Strong performance shown in the experiment section, although the paper mainly focuses on LaMAR HGE dataset

(+) Compared to the previous method, such as $F^3Loc$, the proposed method is more general, in the sense that it does not rely on per-scene depth estimator, which significantly improves the usability of such methods

(+) the writing is the paper is good -- it is relatively easy to the reviewer to follow

(+) Using a depth encoder seems new to me and makes a lot of sense

**Weaknesses:**

(-) Biggest concern from me is the lack of novelty and suitability in the sense of the ICLR (learning representation) community. The paper is a 3D computer vision problem in pose estimation in a room. The main contributions / problems to improve from previous methods are 1/ the lack of uncertainty modeling in depth predictions and 2/ the need for custom, environment-specific depth networks for each environment. These are very specific novelties for this particular task, instead of any major novelty for the learning representation community

(-) The proposed method feels like using / combining a few existing tools/engineering techniques (DepthAnything, Rotate/Translate the images, Floorplan matching, and heavy post-processing etc) and made it work for a particular task. This feels more suitable for submission to a 3DV conference, CVPR or robotics conference, instead of a machine learning conference

(-) The performance of the method will be limited by DepthAnything's accuracy, so this paper essentially trades accuracy ceiling with generalizability

(-) the experiments focus heavily on a small dataset

**Questions:**

n/a

---

> ### Author Response · Authors · 2025-11-22
>
> We thank the reviewer for acknowledging our “strong performance”, improved generalizability over prior work, “good writing”, and recognizing that using a depth encoder “seems new and makes a lot of sense”.
>
> ### Suitability for ICLR
> We respectfully disagree that our work is unsuitable for ICLR. Our contributions align directly with the conference’s scope (see “About Us” on https://iclr.cc/):
> - “Representation learning for computer vision”: We demonstrate that representations learned for depth estimation tasks transfer effectively to floorplan localization. To our knowledge, this is a previously unexplored connection. Table 5 shows that mono-depth pretrained encoders (Depth Anything v2) significantly outperform general-purpose vision encoders (DINOv2) for this geometric reasoning task (+16.8 percentage points when averaged over all sequences lengths).
> - “Applications in robotics”: Indoor localization is fundamental to (mobile) robotics, which is explicitly listed in ICLR’s scope
> - Previous publications at ICLR: Vision-geometry papers regularly appear at ICLR (e.g., MagicDrive, DeepSPF, Hand-Object Dynamics (all ICLR 2024))
> - Our contribution: While primarily methodological and empirical rather than theoretical, we demonstrate how to effectively adapt frozen foundation models for geometric tasks requiring uncertainty. Our framework achieves 42x improvement on short sequences (LaMAR HGE) and significantly improves zero-shot transfer (0% to 50% on new buildings, Table 4). The 24.1-point average improvement across all sequence lengths and the 41.2-point improvement on short sequences (T=15) on LaMAR HGE validate these design choices matter beyond encoder selection. These insights apply to other geometric reasoning tasks such as SLAM, tracking, visual odometry.
>
> ### “Engineering” vs. research
> The large improvements (42x on short sequences, 2.7x on long sequences) indicate more than incremental engineering:
> 1. Novel uncertainty-aware framework: We formulate the first probabilistic model predicting both floorplan depth and uncertainty (Eq. 5), which modulates the localization process via Laplace distributions (Eq. 1) and guides post-processing (Eq. 8).
> 2. Non-obvious design requiring research insight: Prior work (F3Loc, LaLaLoc, LASER) used ImageNet ResNet-50, suggesting general encoders were sufficient. Our hypothesis that task-aligned depth encoders would outperform required systematic investigation across encoder families (Table 5).
>
> ### Performance limited by accuracy of Depth Anything
> It is true that the performance is limited by the features of the pre-trained encoder. However, note that previous work did rely on pre-trained encoders as well (e.g., F3Loc, LaLaLoc, LASER), mostly on ResNet encoders. Training also the encoders from scratch (not just the floorplan depth prediction) would likely increase the requirements regarding data quantity and make training much more difficult in general (since the model would first have to learn basic vision features). Another option could be to fine-tune the pre-trained encoder, which we however did not look into.
> Using off-the-shelf depth models and integrating them with uncertainty modeling, we show that:
> - Current ceiling is high: With Depth Anything v2, UnLoc achieves 97.3% SR on 100-frame sequences and 81.3% on 15-frame sequences (LaMAR HGE)
> - Generalizability is stronger: F3Loc fails to generalize across environments (0%), while UnLoc achieves 50% SR on unseen scenes in Table 4.
>
> An additional important point is that using pre-trained mono-depth encoders brings future-proof modularity. As depth models improve (e.g., Table 5 shows gains from Small to Base to Large models), our method automatically benefits.
>
> Regarding the “heavy post-processing” concern: Our uncertainty-weighted refinement (Eq. 8) adds only 0.96s per sequence. It is lightweight optimization that improves accuracy at a small computational cost.
>
> ### Experiments Focus on Small Dataset
> We evaluate across four datasets with substantially diverse characteristics:
>
> |Dataset|Type|Scene scale|Dataset size|Characteristics|
> |-|-|-|-|-|
> | Gibson|Synthetic|188 small-scale scenes (<300 m²)|>20k training images|Upright poses, low-medium occlusion, large FoV (108°)|
> |LaMAR HGE|Real-world|single large-scale scene (22500 m²)|3820 training images|Non-upright poses, low-high occlusion, narrow FoV (48°)|
> |LaMAR CAB|Real-world (cross-domain)|single mid-scale scene (22500 m²)|329 training images (only used for fine-tuning)|Non-upright poses, low-medium occlusion, narrow FoV (48°)|
> |Structured3D|Synthetic|3296 small-scale scenes|78453 images in total|Non-upright poses, low-medium occlusion, medium FoV (80°)|
>
> LaMAR HGE is emphasized because it represents the most challenging real-world conditions (large-scale, narrow FoV, high occlusion, non-upright poses) where prior methods fail dramatically (e.g., F3Loc: 1.2% on short sequences). This is where our contributions matter most.

---

### Official Review · Reviewer_wgVS · 2025-11-01

**Soundness:** 3
**Presentation:** 3
**Contribution:** 2
**Rating:** 4
**Confidence:** 4

**Summary:**

The paper proposes a method for visual floorplan localization that integrates uncertainty-aware monocular depth estimation. Specifically, the authors leverage large-scale pre-trained monocular depth models (e.g., Depth Anything) to (1) enhance generalization to unseen environments, and further (2) incorporate weighting on depth predictions based on their estimated confidence/reliability. Experimental results show substantial gains in both accuracy and robustness over existing methods.

**Strengths:**

- Simple yet sound idea that leads to performance improvements.
- This presentation is clear and easy to follow, with figures that are mostly intuitive and effectively support the explanations.
- The evaluation section is thorough – Table 1 and 2 showing sequence localization on Gibson and LaMAR HGE datasets, respectively. The author also reports the runtime which is crucial for localization tasks. Furthermore, the paper also ablates different depth estimation models and validates the effectiveness of the proposed uncertainty estimation.

**Weaknesses:**

- The overall technical contribution is somewhat limited, as the main novelty lies in replacing the F3Loc depth networks with SOTA monocular depth estimation models. Moreover, uncertainty modeling in depth estimation has already been explored in many prior works [1, 2].
- Lack of discussion about the uncertainty in the related works or preliminary. The author simply throws the equation at Sec3.5 without sufficient explanation of its derivation or references to prior studies that motivated it.

[1] Kendall and Garl, What Uncertainties Do We Need in Bayesian Deep Learning for Computer Vision?

[2] Poggi et al., On the uncertainty of self-supervised monocular depth estimation.

**Questions:**

- Is this depth encoder being fine-tuned during training? If it is, wouldn’t this make the approach similar to F3Loc, which requires model training for each environment?

---

> ### Author Response · Authors · 2025-11-22
>
> We thank the reviewer for the careful assessment and for acknowledging our work as “simple yet sound” with “thorough” evaluation and “clear” presentation.
>
> ### Technical contribution
> We respectfully disagree that our contribution is primarily a replacement of the depth network. The performance gap between “F3Loc + Depth Anything v2” and our full method shows that improved depth alone cannot account for the results. Averaged over all sequence lengths, our method improves SR@1m from 75.6% to 90.6% on Gibson, 48.6% to 72.7% on LaMAR HGE (Original), and 57.4% to 83.4% on LaMAR HGE (Cropped) compared to “F3Loc + Depth Anything v2”. These gains stem directly from the additional modeling we introduce: an uncertainty-aware observation model tailored to floorplan localization, uncertainty-weighted sequential fusion, and a lightweight refinement stage with 0.96s average runtime. Without these components, depth replacement alone closes only a fraction of the gap, particularly at shorter sequence lengths, where the improvement is most pronounced.
>
> ### Relation to prior work on depth uncertainty
> We appreciate the reviewer’s pointers to relevant works on prior uncertainty-aware depth estimation. We have extended our discussion on depth uncertainty, integrating the two references (see new version of paper).
> To clarify the distinction: These studies focus on the uncertainty of dense monocular depth predictions themselves. Our work models floorplan depth uncertainty and incorporates it explicitly as an observation likelihood within a histogram-based localization framework. In short, we do not only estimate uncertainty: We also introduce the task-specific components, such as uncertainty-aware matching and post-processing, that allow to actively leverage it for localization.
>
> ### Uncertainty Discussion
> We appreciate this feedback. We have significantly expanded our discussion in the revision. In Section 2, we enhanced the “Depth Uncertainty” paragraph to distinguish between aleatoric and epistemic uncertainty, discuss prior work [1,2], and explain our focus on aleatoric uncertainty (inherent observation noise from scene ambiguities). In Section 3.5, we clarified that the Laplace distribution enables closed-form likelihood computation for efficient histogram filtering while providing robustness to outliers due to its heavy tails. The choice of the Laplace distribution is supported both empirically (Table 11) and theoretically, motivated by the need for robustness in indoor scenes with challenging elements such as glass, doorways, and occlusions.
>
> ### Q: Is the depth encoder fine-tuned?
> No. The depth encoder is frozen. Only the masked-attention module is trained. This contrasts with F3Loc, which must learn floorplan-specific depth prediction from scratch. Because our backbone already encodes geometry learned at scale, UnLoc transfers significantly better to new environments. For example, when trained on LaMAR HGE and evaluated on the unseen LaMAR CAB building (Table 7), UnLoc achieves SR@2m of 100% / 40% / 57.1% / 30.8% / 16.7% for sequence lengths T=100 / 50 / 35 / 20 / 15, whereas F3Loc fails completely (0% across all T).

---

### Official Review · Reviewer_wqUb · 2025-11-02

**Soundness:** 3
**Presentation:** 3
**Contribution:** 2
**Rating:** 6
**Confidence:** 2

**Summary:**

The paper presents a method for floor plan localisation based on a sequence of input images. Its main contribution lies in introducing an uncertainty-aware depth estimation framework and replacing customised depth models with modern monocular depth networks. As a result, the proposed approach outperforms prior methods on two datasets, LaMAR and Gibson.

Strength: The idea of incorporating uncertainty into the depth matching process is novel and technically well-motivated. The experimental results convincingly support the method’s effectiveness.

Weakness: The evaluation lacks results on the Structured3D dataset, which is used by the closely related baseline F3Loc. Moreover, replacing a customised depth network with a modern monocular model, while beneficial, is a relatively straightforward modification.

**Strengths:**

1. The idea of introducing uncertainty into the depth matching process is novel and technically sound.
2. The experiments are comprehensive and largely convincing.
3. The paper is well written and easy to follow.

**Weaknesses:**

1. The results on the Structured3D dataset are missing, which is used in the closely related baseline F3Loc.
2. Replacing the customized depth networks with modern monocular depth networks is relatively straightforward.

**Questions:**

1. Could you include results on the Structured3D dataset?
2. Are there any challenges in replacing the customized depth network with monocular depth networks?

---

> ### Author Response · Authors · 2025-11-22
>
> We thank the reviewer for highlighting the novelty of our integration of uncertainty into the depth matching process, the technical soundness of our approach, and the quality of our experimental evaluation.
>
>
> ## Missing Structured3D results
> We appreciate the reviewer raising this point. Structured3D results are provided in Table 6 (Appendix A.2.1) for single-frame localization. Our focus, however, is sequential localization, which we view as the most relevant setting for robotics and AR. Structured3D contains only independent images and does not provide trajectories, making sequential evaluation impossible. We therefore focused our sequential evaluation on Gibson and LaMAR, which do contain sequences.
>
>
> ## “Straightforward replacement” of the depth network / Challenges
> While implementation of the replacement is simple, the design choice itself was not obvious. Prior work (F3Loc, LaLaLoc, LASER) relied on ImageNet-pretrained ResNet-50, suggesting that established general-purpose encoders were sufficient for this task. We hypothesized that task alignment matters: Depth encoders trained on scene-to-depth mapping would better capture geometric priors than classification encoders.
> Empirical validation (Table 5) confirmed this hypothesis: DepthPro (L) achieves on average +12.1 and Depth Anything V2 +16.8 percentage points in SR@1m compared to Dinov2. Even smaller mono-depth encoders (Depth Anything V2 (B/S)) outperform large general encoders once uncertainty is incorporated. To our knowledge, this is the first demonstration that monocular depth encoders transfer more effectively than generic vision backbones for floorplan-based localization, providing insight into how task-aligned representations benefit geometric reasoning tasks.
>
>
> ## Challenges in replacing the customized depth network
> Several challenges emerged. First, different encoder architectures produce features with varying dimensions, requiring careful adaptation to match the masked attention's spatial resolution requirements. We addressed this through bilinear interpolation. Second, we had to verify empirically that representations learned for scene depth (camera-to-object distances) would transfer effectively to the different task of predicting floorplan depth (distance to nearest wall), which was not guaranteed a priori. Third, the off-the-shelf depth encoders don't directly provide the uncertainty estimates needed for our floorplan depth predictions, necessitating the design of our dual-head architecture that learns to predict both floorplan depth and associated uncertainties from encoder features. Finally, we needed to preserve the geometric priors learned during pretraining on large-scale depth datasets, which we achieved by keeping encoders frozen while learning only lightweight prediction heads.

---

> ### Comment · Reviewer_wqUb · 2025-11-27
>
> I appreciate the detailed author response. It appears that all reviewers share a central concern regarding the integration of a modern depth network into the floor plan localisation pipeline. Although this is largely an engineering task, the experimental results are encouraging. After considering all reviews and rebuttals, I still lean toward a weak accept, though I would also be comfortable with the work being recommended for submission to a future venue.

---

### Meta-Review · Area_Chair_VWUJ · 2026-01-03

**Summary:**

The paper initially received mixed reviews with scores of 6, 4, 4, and 2.
The reviewers raised concerns about the paper's novelty and technical contributions. Their primary argument is that the paper's innovation comes mainly from substituting the F3Loc depth networks with state-of-the-art monocular depth estimation models. Some reviewers noted that the paper mainly integrates existing methods and applies them to floorplan localization, which they consider a limitation in its scope.

The authors' rebuttal effectively addressed the reviewers' concerns by clarifying why the proposed method is not merely a direct replacement for the depth network and outlining potential challenges in substituting the customized depth network. Additionally, the experimental results obtained from the Gibson and LaMAR HGE datasets supported the authors' claims. The authors also provided new results on the Structured3D dataset in the Appendix as requested by the reviewers.

The area chair has reviewed the paper along with the reviewers' comments and has confirmed that the authors have adequately addressed most of the raised concerns. Additionally, this paper successfully improves previous work in two important aspects: i) by incorporating uncertainty modeling, and ii) by removing the need for dataset-specific depth. These contributions are noteworthy. Therefore, the area chair recommends accepting this paper.

**Reviewer Concerns:**

As mentioned in the summary, the authors have addressed most of the reviewers' concerns.

**Reviewer Scores:**

According to the comment from **Reviewer wqUb**, the area chair anticipates that the reviewer will keep the initial score at 6.
> After considering all reviews and rebuttals, I still lean toward a weak accept, though I would also be comfortable with the work being recommended for submission to a future venue.

**Reviewer wgVS**, **Reviewer Lj7y**, and **Reviewer PYiR** did not provide further responses.
However, **Reviewer PYiR** mentioned in the initial review that
> If the authors could provide a more comprehensive description of the evaluation methodology and experimental setup on LaMAR, I would consider improving my overall rating.

Since most of their concerns have been resolved, the area chair expects that **Reviewer wgVS**, **Reviewer Lj7y**, and **Reviewer PYiR** will likely improve their scores.

The final scores are expected to be **6**, **6**, **6**, and **4**.

---

### Decision · Program_Chairs · 2026-01-26

Accept (Poster)